# Exploring battery usage in Electric Vehicles through Graph based Cascaded Clustering

## Abstract

Recently, electric vehicles (EVs) have gained popularity over internal-combustion engine vehicles (ICEV) because of their convenience and ability to use clean-energy sources. Fueling an EV is fundamentally different than an ICEV and thus, the driving and charging patterns associated with EVs are novel and not well understood. For example, where filling the tank of an IECV is a standard process, charging an EV occurs at different speeds (L1, L2, and DC Fast Charging) and intermittently while driving by regenerative braking. Understanding these usage patterns for EVs is important because the performance and longevity of the battery is dependent on the driving, charging, and idling patterns it is subjected to over its lifetime. We propose a scalable cascaded clustering approach that leverages battery-specific features to identify usage patterns that affect the battery across multiple timescales. We analyze 3,100 EVs over the course of a year using multivariate time-series data consisting of but not limited to state of charge (SOC) and mileage. First, we apply clustering to weekly multivariate data segments and extract usage profiles at that timescale. Then, we use these weekly cluster assignments to generate an EV battery meta-sequence that is unique to every vehicle, which reveals longer-term patterns. We apply a novel graph based clustering technique at the vehicle meta-sequence level to associate groups of vehicles that are operated similarly. Our approach reveals fine-grained usage patterns and helps identify salient themes across a vehicle's lifetime. While limited to a relatively small selection of vehicles, our work reveals a unique representation of vehicles and their weekly usage pattern that can potentially aid in battery lifecycle management.

## 1 Introduction

In the realm of electric vehicles (EVs), a significant knowledge gap exists regarding the comprehensive understanding of EV and battery usage, with the first crucial step being an in-depth examination of EV utilization, encompassing charging, discharging, power consumption, and idle patterns. The incorporation of sensors and communication technologies in EVs has opened avenues for enhanced insights into EV battery usage, enabled by time-series data analysis and clustering techniques. Our goal is to cluster similar vehicles based on shared usage tendencies and also explore general weekly EV usage patterns across different vehicles. Our in-house dataset consists of 3100 EVs with approximately 1.5 years of data recorded. We build a cascaded clustering framework that involves two different stages of clustering: the first stage gives us an insight into weekly usage patterns across all vehicle-agnostic weeks, while the second stage of clustering is used to cluster entire vehicle usage patterns using meta-sequences from stage-1 clustering. This approach enables a comprehensive understanding of long-term themes and collective EV weekly behavior, potentially informing energy optimization, demand forecasting (using our deep embedding), and grid management decisions in the future, although this is out of the scope of our paper. The main contributions of our paper are two-fold:

1. We derive domain specific features that produce insightful EV weekly-usage patterns

2. We propose a novel deep clustering network combining a Long Short-Term Memory architecture (LSTM) and a Graph Convolutional Network (GCN) to analyze vehicle meta-sequences

The paper Nazari et al. (2023) presents an overview of the different clustering methods applied to different types of EV data. Some of the most popular studied application areas are EV user behavior in Helmus et al. (2020); Hu et al. (2022); Powell et al. (2022), EV driving cycle in Berzi et al. (2016);

Zhao et al. (2019), the classification of used EV batteries in Li et al. (2022; 2019a); Liu (2019), and clustering of EV charging stations in Momtazpour et al. (2012); Sánchez et al. (2022). In contrast to the methods discussed in Nazari et al. (2023) pertaining to EV user behavior, we employ a novel cascaded clustering approach on multivariate time-series data as mentioned in Section 2.2. Similarly, the approach proposed by Xiong et al. (2018), which combined K-means clustering and a multilayer perceptron to analyze historical electric vehicle charging station data with the goal of identifying station usage patterns differs from ours. We aim to extract both driving and charging patterns directly from individual vehicle data over an extended period. Moreover, our approach differs in that it employs a cascaded clustering technique, eliminating the need for manual labeling by leveraging the clustering labels generated in the initial step. A type of cascaded clustering was introduced by Yildiz et al. (2021). It differs from our approach in two ways. It initially clusters data at hourly and daily levels, followed by a re-clustering step to consolidate similar clusters and refine each one. Our method, in contrast, starts with granular weekly patterns and progresses to higher-level clustering based on vehicle-specific labels assigned weekly, resulting in a more generalized approach compared to fine-grained clustering.

Our approach draws inspiration from Li et al. (2019b), but it distinguishes itself through significant enhancements. In Li et al. (2019b), a two-stage clustering methodology is proposed, beginning with the clustering of daily electric vehicle (EV) data to uncover common driving patterns on different days. Subsequently, they use these daily patterns to calculate a distribution that quantifies their occurrence relative to the total number of travel days for the EV. Finally, K-medoids clustering is applied to group these diverse driving patterns. Our approach enhances the second-stage clustering by incorporating an LSTM layer with a GCN for fine-grained temporal relationships, and adopts weekly data segments in the first-stage clustering, contrasting with the daily segmentation in Li et al. (2019b). This enhancement results in superior performance on our dataset.

Graph neural networks (GNN) have become very popular in recent years, recognized for their ability to efficiently and accurately perform regression, classification, and even clustering tasks on graph-structured data. Bianchi et al. (2020) details one of the most popular methods for graph clustering, known as MinCutPool. Tsitsulin et al. (2020) improved upon this approach with their Deep Modularity Network (DMoN), which made use of a differentiable form of the modularity equation to directly learn cluster labels with a graph neural network in an unsupervised fashion. The application of GNN's to time series tasks has also been explored. Cini et al. (2023) leverages clustering and GNNs to learn the hierarchical relationships between time series features, and applies this to a supervised forecasting problem. Similarly, Wu et al. (2022) leverages k-shape clustering to determine usage profiles before leveraging a GNN framework to learn a time-series forecasting problem. Zhu et al. (2023) leverages GNN's for multivariate time series clustering, modeling each variable as a node and analyzing their relationship, claiming state-of-the art results on several time-series classification datasets.

The remainder of this paper is organized as follows. In Section 2, our Cascaded clustering framework is presented along with the data description and preprocessing. The experimental results of our cascaded clustering, relevant usage patterns discussions are provided in Section 3. Limitations and next steps are giving in Section 4. Lastly, the conclusions are given in Section 5 and the additional materials in the Appendix A.

## 2 FRAMEWORK AND APPROACH

We build a scalable Cascaded Clustering framework as shown in Figure 1. This framework enables us to group the EVs that overall exhibit the most similar patterns in charging and driving behavior. All our experiments are executed on a 12-core CPU MAC M2.

### 2.1 DATA PREPROCESSING

The data used in this study is collected over-the-air from EVs in the United States. There are 3,100 individual vehicles included in the dataset, identified through anonymized VINs (vehicle identification number). The number of measurements for each VIN varies, but in total there are over 3.8 million measurements. Each measurement has 5 features of interest, including the timestamp, an anonymized VIN, a boolean flag indicating whether the vehicle is at 'home' or not, the odometer reading, and SOC%. The battery state of charge is expressed as a percentage of fully charged capacity. Figure 2 summarizes the dataset as well as some of the key features. The number of data samples per VIN

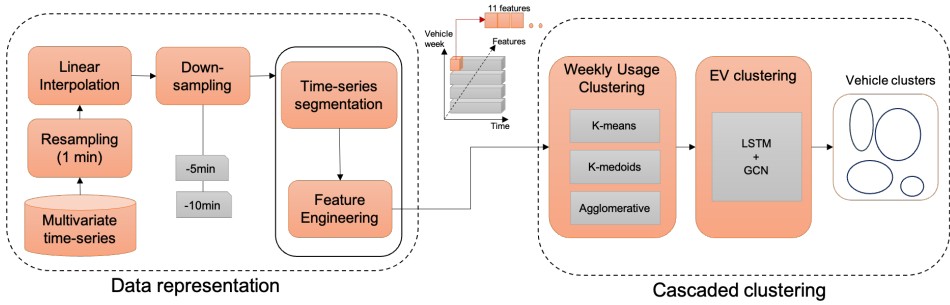

Figure 1: cascaded clustering framework

varies greatly, so the total measurements per VIN are plotted as a histogram in plot 2a. Most vehicles have fewer than 6,000 recordings, but some have as many as 16,000. Plot 2b shows the idle, charging, and driving count of measurements for 100 selected vehicles. We plot mileage and SOC% for 5 vehicles and 1 vehicle respectively in Appendix A.1

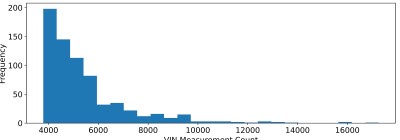

(a) Histogram of Total Counts for Each Vin

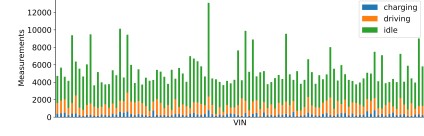

(b) Driving, Charging, and Idle Measurement Breakdown for 100 Vins

Figure 2: Summary of raw data characteristics and visualization

Our data is a multivariate time series, where each datapoint comprises both a feature axis and a time axis as shown in Figure 1. Notably, the time-steps across the datapoints do not align due to varying or missing recording times for different vehicles, albeit within the same overall timeframe. To standardize the time-scale and address missing data, we employ resampling and linear interpolation techniques. First, we resample our data to a higher frequency (1 min) to capture intricate details and then perform linear interpolation on all features . Next, we downsample our data to 10 min frequency in order to save on computation. Considering that battery usage patterns unfold gradually over extended time intervals, a lower sampling frequency, such as 10 min, suffices for our analysis. To handle missing data, we identify and flag instances of missing values. Additionally, we remove data chunks with missing values exceeding a duration of one day. Also, we exclude vehicles from the analysis if they possess an insufficient number of recordings.

## 2.2 FEATURE ENGINEERING AND TIME SERIES SEGMENTATION

In addition to the signals included with the raw data, we derive domain-specific features that characterize battery usage with respect to power, energy and driving/charging events. Some of the features include $\Delta$SOC, an event index, depth of discharge, charging energy, and charging power. We start with $\Delta$SOC which is simply the difference in subsequent measurements of SOC. This helps to characterize both charge and discharge power on a continuous basis. We derive an event index to keep track of different charge and discharge cycles. For example, a charging event followed by a discharge (driving) event would be two events. We use a threshold of 2% as a minimum where any changes less than that would not be counted as a new event. We use a 2% SOC change threshold within a 10-minute window to distinguish charging and driving events. This threshold serves two key purposes. Firstly, it filters out small, irrelevant charging events like brief plug-and-unplug actions. Secondly, regenerative braking occurs during driving, converting motion energy into electrical energy. To avoid counting regenerative braking as charging, we use the SOC threshold. Regenerative braking's SOC increase in electric vehicles varies due to factors like design, battery capacity, and driving conditions. We explored our dataset and opted for a 2% SOC threshold. Next, depth of discharge (DOD) is calculated as the total change in SOC during a discharge cycle. For charging energy, we simply use

the DOD for charging events and multiply by the rated nameplate capacity of the battery, $E_{rated}$. This is given by equation 1

$$E_{charging}(kWh) = DOD * E_{rated} \tag{1}$$

For power, we derive a three stage charging power classification that reflects the maximum charging rate used during a charging session. These are categorical values $\in$ {slow, fast, rapid} with the following thresholds: slow $< 0.054 * \frac{SOC\%}{min} \leq$ fast $< 0.417 * \frac{SOC\%}{min} \leq$ rapid. These thresholds have been converted to SOC/min for consistency with our datasets, but they can also be expressed in kW, as 2.8kW and 21.5kW respectively. Apart from the above mentioned features, we also derive velocity, overall event (charge/discharge) number, weekly event number, total weekly mileage, rate of change in energy, rate of change in mileage. This helps in extracting patterns that might not come through from the original data and aids in post processing cluster descriptions. In total, we have 11 features describing every vehicle at 10 min intervals over $\sim$ 1 year. We explain all the other features not described in this section in Appendix A Section A.2 in order to aid reproducibility.

## 2.3 CASCADED CLUSTERING

In our academic investigation, we face a crucial decision at stage-one on how to cluster structured weekly time-series data. While Dynamic Time Warping (DTW) with K-means Niennattrakul & Ratanamahatana (2007) is effective for variable-length time series, our preprocessing steps, like interpolation and segmentation, impact DTW's suitability. Preserving the temporal aspect is paramount, especially for metrics like State of Charge (SOC%) in rapid charging scenarios. To avoid temporal distortion, we exclude DTW due to its warping nature. Given our large dataset (21,000 weeks, 1,008 measurements per week, 11 features per measurement), raw clustering isn't computationally feasible. We briefly explore the viable strategy of dimensionality reduction (e.g., UMAP) before applying a clustering algorithm. Another popular technique is to apply clustering on pre-computed distance matrices (Holder et al., 2023). We run experiments using distance matrices with K-means Lloyd (1982), K-medoids Park & Jun (2009), and Agglomerative clustering Patel et al. (2015). These approaches aim to efficiently extract insights from our complex temporal data.

In the context of clustering noisy and outlier-rich data, the choice between Manhattan distance (L1) and Euclidean distance (L2) becomes pivotal. Manhattan distance, computed by summing absolute differences along each dimension, offers robustness to outliers as it considers the magnitude of differences rather than their squared values. This makes it a preferred choice for clustering in noisy environments. In contrast, Euclidean distance based on squared differences, is sensitive to outliers, potentially leading to less reliable clustering outcomes in noisy data scenarios. In Kobylin & Lyashenko (2020) a similar type of segmentation of heartbeat data is performed and the observed results for K-means with Euclidean distance and Soft-DTW with K-means were almost identical. We apply the K-means technique with Euclidean distance, but also consider K-medoids since K-medoids is known to work better with noisy data since its algorithm centers around medians and not means. Agglomerative clustering can be preferable for multivariate time series data due to its flexibility in capturing variable cluster shapes, its ability to provide a hierarchical structure for exploration, and its robustness to outliers and initialization compared to K-means or K-medoids. Since there is no "universal" best option in clustering data, we choose to experiment with the three above mentioned techniques. While performing experiments with K-means and K-medoids, we identify the optimal number of clusters by using a combination of the elbow method, Silhouette score and Davies-Bouldin index. This is done by varying the number of clusters for distance metrics such as Euclidean and Manhattan respectively and recording the scores for each value of K as shown in the Appendix A, Figure 2a. The Silhouette score in blue should be maximized, the inflection point of inertia in red should be found, and Davies-Bouldin index in yellow should be minimized. For Agglomerative clustering, we specify a distance threshold based on our choice of distance metric, and allow the algorithm to converge at a specific $K_{stage1}$. We also use a dendogram for the same to aid in narrowing the possibilities of optimal clusters K at stage-one, $K_{stage1}$.

In Stage-2 of our cascaded clustering framework, we introduce a novel technique for deep clustering as shown in Figure 3. As the first step, we use a time series encoder. This layer serves to create the node representation for each time series sequence of cluster labels. Since the LSTM is a learnable layer, the model will be able to update the encodings throughout the training process, adapting the node representation to contain the information most relevant to the separation of time series

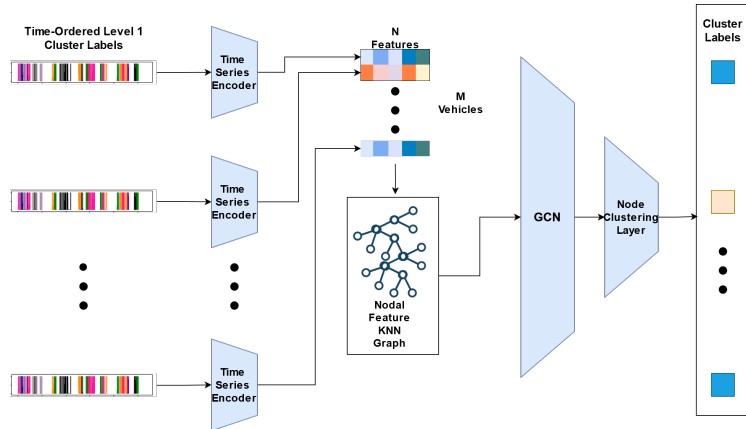

Figure 3: LSTM encoder + GCN technique for Stage-2 clustering

subsequences into clusters, while maintaining the time-ordered information. This layer will convert the M meta-sequences intoan M × N matrix, where N is the number of features output from the time series encoder. The time series encoder is comprised of an LSTM auto-encoder, which have been proven to be very effective at analyzing and extracting data from time series. This model architecture is shown above using vehicle time series sequences of stage 1 labels as an input. The nodal feature adjacency matrix defines the initial edges between the node embeddings from the time series encoder. The graph representation is constructed using a K-nearest neighbor graph based on the embeddings output from the LSTM encoder. The graph convolution layer enables the analysis this graph-structured data in a learnable neural network. While the initial graph representation acts as a seed, this layer can learn to modify the node representations in order to create a more effective representation for clustering.

Finally, the clustering layer takes the learned graph representation as an input, and outputs a cluster label for each node (each meta-sequence). The loss function seeks to maximize modularity, which is the same goal of many of the naïve community finding algorithms, such as the Louvain algorithm first proposed in Blondel et al. (2008). Modularity is a metric of scoring community detection in graphs, and graphs with high modularity feature dense connections within their community, with fewer connection between communities. Community detection is similar to clustering, but on graph-structured data, where a community of nodes is analogous to a cluster of data. Modularity is not differentiable, however, implementing a spectral relaxation of modularity as a differentiable function, as outlined in Tsitsulin et al. (2020), enables end-to-end learning, where both the time order-informed LSTM embedding and cluster labels can be jointly learned to optimize the final clusters.

Our results from both stages of clustering are summarized in Section 3.1.

## 3 RESULTS AND DISCUSSION

### 3.1 EXPERIMENTAL RESULTS

We have summarized our key results after performing stage-one clustering on our multivariate data into Table 1. For our dataset we find that both K-medoids and K-means have similar scores and we run experiments through our pipeline on both methods. K-medoids works best for $K_{stage1} = 6$ using Manhattan distance as it is able to provide cleaner usage profiles as shown in Figure 4. We see a clear difference in usage-profiles when these results are compared to the ones produced by K-means, as shown in Appendix A, Figure 5. Since a medoid is always a specific datapoint from the dataset, it is less affected by outliers than the mean (K-means' centroid) and seems to give our usage profiles more structure. The poor performance of Agglomerative clustering could be due to poor estimation of distance threshold, or badly interpolated values in the data. We have summarized a set of relevant experiments (including random seed error report) for stage-one clustering in Section A.3 in Appendix A. In order to show the robustness of our engineered features, we run a comparative experiment showcasing Stage-One clustering with just a basic set of the collected features. We see that our features improve validation metrics for both K-means and K-medoids. This is shown in

Table 3. We observe that implementing dimensionality reduction using UMAP McInnes & Healy (2018) results in improved time efficiency for clustering, along with higher Silhouette scores and lower Davies-Bouldin indices. However, when visually inspecting the clustered projections in both scenarios, it becomes evident that dimensionality reduction has failed to capture meaningful clusters in our specific case. This is shown in Appendix A, Figure 7 where dimensionality reduction before clustering simply splits the data into clusters without any meaningful grouping while this isn't the case when we cluster on raw data.

The paper Li et al. (2019b) proposes a two-stage clustering technique and we use it as a baseline on our dataset (listed as Stats K-means). From Table 2, we see that our approach outperforms this baseline method of using meta-sequence compositions as input at the second stage of clustering. Additionally, we compare our approach to clustering on the raw meta-sequence labels as well as a learned LSTM embedding. The LSTM embedding is generated using a simple LSTM autoencoder architecture trained to minimize reconstruction loss. This differs from our LSTM + graph convolution network approach, which is trained to minimze both the reconstruction loss and clustering loss. From the results in Table 2, it is clear that our method outperforms the baseline methods, including the LSTM embedding, likely due to the use of the clustering loss to improve the LSTM embedding. Apart from using an LSTM, we also ran an experiment using a Transformer followed by a GCN, but this model didn't converge. We suspect that this is due to not having enough data after eliminating idle weeks from the meta-sequences.

It should be noted that in the second stage of clustering the approaches are evaluated on 2 different subsets of our original dataset. Due to a large number of idle weeks degrading model training, two thresholds were set to elimnate vehicles with mostly idle weeks. The threshold was set to 10 (out of 63) weeks, yielding a dataset with 785 vehicles, and 20, yielding a dataset with 111 vehicles. The results for both of these datasets, as well as the open source factory power consumption dataset, are given in Table 2. The open source dataset we use is sourced from Braeuer (2020). It consists of the grid power consumption for 30 different factories over the course of a year. In order to validate our approach on an external (non-proprietary) set of data, we processed it in the same method described above. Power consumption and its first derivative were used as features to generate weekly load profiles (included in Appendix Figure 6), which were then used in the second stage cluster approaches listed below. Due to a lack of available datasets featuring 2 stages of labels, the approach could not be evaluated on any open source classification dataset, and is only evaluated using its unsupervised performance metrics.

Table 1: Clustering stage-one results

| CLUSTERING ALGORITHM | SILHOUETTE SCORE | D.B.INDEX | DISTANCE METRIC | OPTIMAL K |
|---|---|---|---|---|
| K-means | 0.054 | 3.036 | Euclidean | 5 |
| K-medoids | 0.035 | 4.029 | Manhattan | 6 |
| Agglomerative | 0.023 | 3.738 | Euclidean | 7 |

Table 2: Clustering stage-two results

| METHOD | SILHOUETTE SCORE | D.B.INDEX | OPTIMAL K |
|---|---|---|---|
| *Number of Vehicles = 111, data = ours, Stage1 Clusters = 6* | | | |
| Stats + K-means | 0.369 | 0.976 | 4 |
| Raw sequence + K-means | 0.0957 | 2.757 | 4 |
| LSTM AE embedding + K-means | 0.425 | 0.819 | 10 |
| **LSTM + Graph Clustering (ours)** | **0.703** | **0.519** | **6** |
| *Number of Vehicles = 785, data= ours, Stage1 Clusters = 6* | | | |
| Stats + K-means | 0.29 | 1.219 | 4 |
| Raw sequence + K-means | 0.066 | 3.297 | 5 |
| LSTM AE embedding + K-means | 0.626 | 0.611 | 4 |
| **LSTM + Graph Clustering (ours)** | **0.63** | **0.527** | **6** |
| *Number of factories = 30, data = Opensource dataset (Braeuer, 2020), Stage1 Clusters = 8* | | | |
| Stats + K-means | 0.742 | **0.252** | 8 |
| Raw sequence + K-means | 0.4257 | 0.73 | 5 |
| LSTM AE embedding + K-means | 0.684663 | 0.3697 | 8 |
| **LSTM + Graph Clustering (ours)** | **0.76** | 0.324 | **4** |

## 3.2 Cascaded Clustering Discussion

**Vehicle Usage Profiles:** While the relative performance of the clustering algorithms was explored in the previous section, we should also examine the battery specific features of each cluster. Figure 4

Table 3: Comparing scores for clustering with and without our derived features on a smaller subset of the data  200 vehicles having the maximum number of measurements

| CLUSTERING ALGORITHM | SILHOUETTE SCORE | D.B.INDEX | OPTIMAL K |
|---|---|---|---|
| *K-medoids* | | | |
| Home, SOC, ΔSOC, weekly mile | 0.07 | 2.74 | 5 |
| All 11 features | **0.082** | **2.688** | 4 |
| *K-means* | | | |
| Home, SOC, ΔSOC, weekly mile | 0.076 | 2.52 | 6 |
| All 11 features | **0.124** | **2.39** | 5 |

shows usage profiles for each stage-one cluster in terms of the raw data signals, SOC, cumulative distance, and home binary. For SOC and cumulative distance, the solid lines represent the median of the cluster selected by the K-medoids algorithm and the translucent lines represent a band of the 200 weeks surrounding those medians. Visibly, all the clusters highlight different usage patterns, though there is still a lot of variability even within the median 200 weeks. In clusters 0, 1, 2, and 5 there are clear trends in the charging pattern. Cluster 0 exemplifies daily usage of the top 60% of SOC. In contrast, cluster 1 is a similar charging pattern with lower depth of discharge and more likely to charge to 100% SOC every evening. Both clusters are indicative of a typical charge-every-night scenario. Cluster 2 shows a different type of behavior where charging is deferred to Thursday nights in particular. Likewise, cluster 5 is a similar but with a preference for Friday nights. In clusters 3 and 4, trends are less clear, however cluster 4 has higher depth of discharge. All clusters show different likelihoods of being home, though no overarching trends are clear. Similarly with cumulative distance, there is a broad distribution in the weekly mileage. While these usage profiles give an overall feel for the median of each cluster, they do not represent the full breadth of the feature set that is used for clustering.

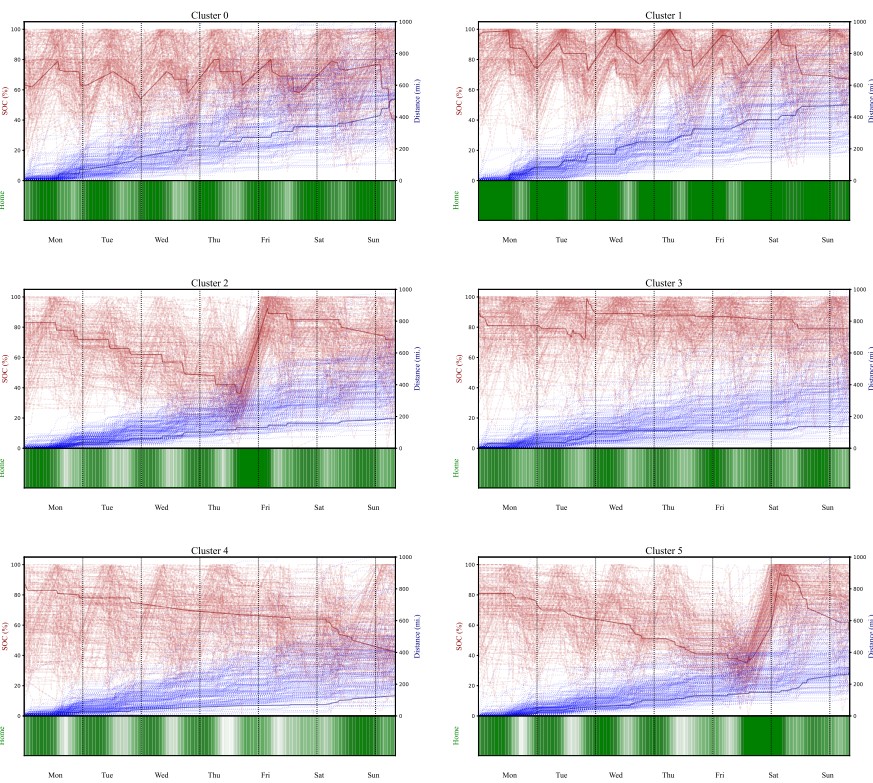

Figure 4: Weekly usage profiles of Clustering Stage 1 clusters for SOC %

**Stage-1 Cluster Behavior and patterns:** In order to better understand the differentiation of our clusters, Figure 5 shows the additional features represented by kernel density estimation (KDE) plots. Figures 5a and 5b show state of charge and depth of discharge (DoD), respectively. Here it is clear to see different utilization of the battery for each cluster. Clusters 0, 1, and 2 show preference to lower

DoD and each has a different likelihood of using an SOC cut off of 80, 90, or 100%. Clusters 3, 4, and 5 however utilize a larger depth of discharge and are more evenly distributed across the full capacity of the battery. These plots show the ability of battery-specific features to separate different utilization patterns. It is important to understand that because of the way KDE plots are calculated, they show a distribution outside the range of possibility. For example, this is the reason why sub figure 5b shows DOD above 100%. This should be ignored as it is not physically possible. Considering sub plot 5c and Figure 6, the charging speed preferences of each cluster is further illustrated. Here the charging duration is a function of the charging power and depth of discharge and we can see the several distributions emerge. Furthermore, Figure 6 shows the relative proportions for Level 1, 2, and 3 charging. By considering all the feature plots, a clearer understanding of the unique characteristics of each cluster emerges. We have included visualizations of meta-sequences tied together in the Appendix A.4

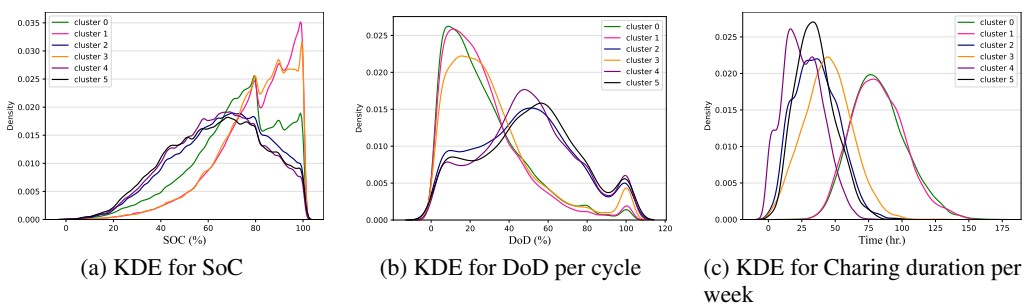

(a) KDE for SoC      (b) KDE for DoD per cycle      (c) KDE for Charing duration per week

Figure 5: Selected features from Stage-1 clustering

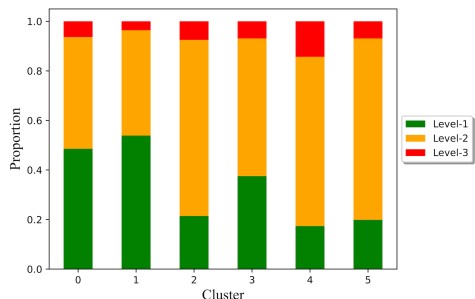

Figure 6: Proportion of different power levels in each cluster with cluster 4 having the maximum amount of fast, high power charging

**Stage-2 Cluster Behavior and patterns:** While the quantitative results for all second stage clustering approaches are given in Table 2, Figure 7 offers a qualitative evaluation of the vehicle-level clustering results generated using our proposed graphical method. Subplots 7a and 7b display heatmaps representing a second stage cluster, with each row representing a different vehicle and each column, a weekly subsequence of that vehicle's life. Each color represents a different stage 1 cluster label. Looking at the figures for clusters 4 and 5, it is clear that the overall composition of these cluster vary greatly, with cluster 7a primarily composed of vehicles with weekly usage profiles 1-2, while cluster 7b features more vehicles comprised of usage profiles 4-5. Additionally, the histograms in subplots 7c and 7d display the overall proportions of each stage-one label in each stage-two cluster.

## 4    LIMITATIONS AND FUTURE STEPS

While the stage one clusters serve their purpose in illustrating our cascaded cluster approach and weekly time segmentation, certain limitations are evident. Our dataset is insufficient in representing all usage patterns, indicating that the "natural" number of clusters is likely greater than the six clusters identified. Consequently, this leads to overlapping cluster metrics. With a more comprehensive dataset encompassing a wider range of usage patterns, we anticipate the emergence of additional

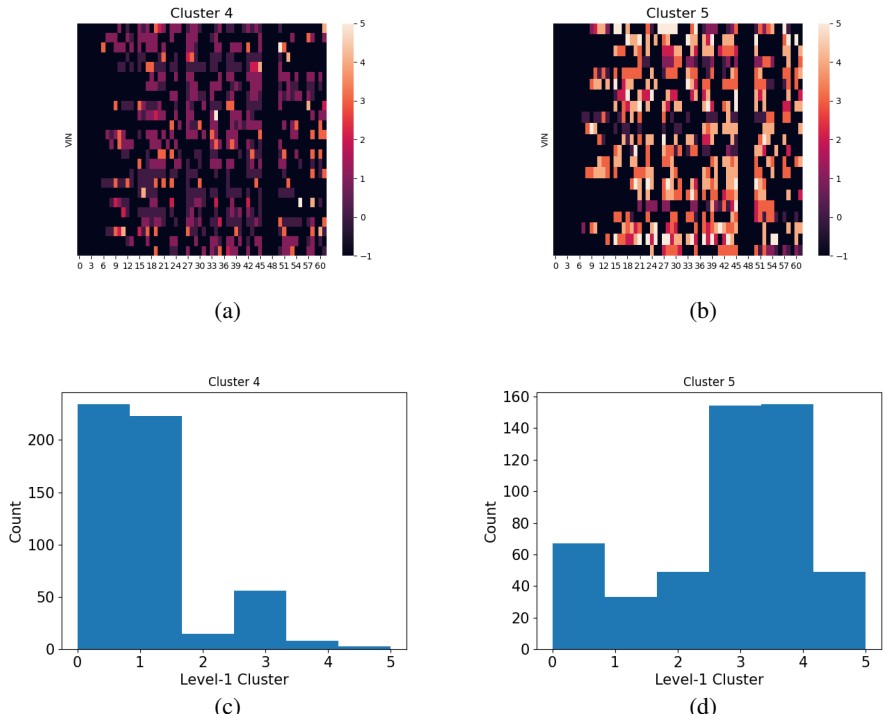

Figure 7: Level-2 cluster distribution insights and heatmaps

distinct clusters. This means that the features employed in the analysis may require further refinement. While clusters 0, 1, and 2 in Figure 4 partially capture variations in preferred cut-off SOC values for charging, there are instances of data overlap among these clusters. Exploring battery-specific features in greater detail is expected to facilitate a better isolation of distinct usage patterns.

Furthermore, our data consists of large and small gaps. Linear interpolation for estimating missing values may generate erroneous patterns in our model, particularly for features such as SOC that require more accurate estimation through advanced modeling techniques. Additionally, applying hard-clustering to our data may have limitations, as real-world EV operation varies over time and clusters can overlap. Furthermore, representing weekly usage profiles by the median value of each feature for a cluster provides a preliminary understanding, but fails to go deeper into explainability and interpretability of the clusters. This is also an area for future work.

Additionally, it should be noted that all nodes need to be present in order to perform graph clustering. Therefore, the proposed architecture must be run with a batch size of one. This may cause memory issues when running on datasets containing a very large number of time series. For very large datasets, it may be beneficial to decouple the time series encoder and graph clustering and instead train them separately. Alternatively, methods of batching the input to the encoder and re-aggregating the mini-batches before passing it through the graph neural network could be explored, and this may be explored in future work.

## 5 CONCLUSION

Our cascaded clustering framework provides valuable insights into the usage patterns of electric vehicles (EVs). At the first stage, we identify short-term usage profiles that reveal how EVs charge and drive on a weekly basis. At the second stage, we cluster vehicle meta sequences to obtain groups of vehicles that have similar cluster-one distributions. Overall, our robust framework enables the processing and understanding of large-scale real-world EV data, which is crucial for facilitating the transition to EVs and anticipating its impacts.

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
