# OpenReview forum: "EXPLORING BATTERY USAGE IN ELECTRIC VEHICLES THROUGH GRAPH BASED CASCADED CLUSTERING"
_ICLR.cc/2024/Conference — ICLR 2024 Conference Withdrawn Submission_

### Official Review · Reviewer_fT58 · 2023-10-21

**Soundness:** 2 fair
**Presentation:** 2 fair
**Contribution:** 2 fair
**Rating:** 3
**Confidence:** 3

**Summary:**

The authors study electric vehicle (EV) usage patterns in the pursuit of providing insights for better battery lifecycle management. Specifically, the authors study an EV dataset consisting of 3,100 vehicles and 3.8 million sample readings each comprised of 5 different measurements such as the timestamp, odometer reading, and state of charge percentage (SOC %). The authors introduce a two-stage cascaded clustering technique to group the data into semantic usage-pattern clusters. Stage 1 consists of applying either K-means, K-medoids, or agglomerative clustering, while stage 2 uses a long short-term memory (LSTM) network in combination with a graph convolution network (GCN) to achieve embeddings that further refine the semantic clusters.

**Strengths:**

* The authors tackle the problem of better understanding usage patterns of EVs. This problem is becoming increasingly important as society shifts to electrifying more and more of the transportation sector.

* Experiments are performed on a relatively large real-world dataset consisting of 3.8 million measurements sampled from 3,100 vehicles.

* The proposed approach is a relatively simple two-stage pipeline in which stage 1 consists of standard clustering techniques and stage 2 uses more complex methods for further refining the clusters.

*  Some of the limitations of the proposed approach are described and discussed.

**Weaknesses:**

* It is unclear to me how meaningful the clusters identified by the proposed cascaded clustering approach are. The insights obtained from using the proposed methodology seem limited, and I'm uncertain of their value (Figure 4). I think a human evaluation of the identified clusters from domain experts would greatly strengthen the experimental evaluation section and any claims about improved insights made in the paper.

* The clarity of the paper could be greatly improved. For example, it's unclear to me how the data is collected; did the authors collect this data themselves, or is this a public dataset? Also, over what time frame does this dataset cover? What is the distribution of vehicle types in the dataset? Please see the "Questions" section for more details about clarity improvements.

* The sample measurements from the vehicles follow a power law distribution, resulting in an experimental evaluation that only depends on a relatively small number of the overall number of vehicles.

* Experiments are only performed using a single dataset, thus, it is unclear how robust these results are. An experimental evaluation using a wider range of datasets would greatly strengthen any generalizability claims made in the paper.

* There are no error bars for the majority of the results (Tables 1, 2, and 3).

* There are grammatical errors throughout the paper, consider using a service like Grammarly to fix these issues.

* I'm unclear of the overall takeaway message of Figure 7, consider adding more clarifying text to the caption and labeling the x-axes in Figures 7a and 7b.

* There is no empirical runtime evaluation of the proposed approach and competing baselines. This evaluation may be very beneficial to readers when deciding whether or not to the use the proposed methodology.

* Figure 2b is not colorblind friendly.

**Questions:**

* How many repeat runs with random seeds did the authors perform for the experimental evaluation?

* Is level 2 cascading necessary? What additional insights does stage 2 clustering provide?

* Have the authors tried using their approach on a non-resampled version of the dataset? If so, have the results differed from using the resampled version of the dataset?

* Why did the authors choose 2% as the SOC % threshold? How did the authors "explore the dataset" to come to this conclusion?

* Why are only 100 vehicles shown in Figure 2b? Are these vehicles sampled uniformly at random?

* How are the 2.8kW and 21.5kW thresholds determined?

* Is higher or lower better for the Silhouette and D.B. Index scores?

* How do the authors define a "meaningful grouping" in regards to clusters?

* How is $k$ selected for the experimental evaluation?

---

### Official Review · Reviewer_kfKg · 2023-10-30

**Soundness:** 1 poor
**Presentation:** 2 fair
**Contribution:** 1 poor
**Rating:** 1
**Confidence:** 5

**Summary:**

The proposed work applies a graph-based clustering algorithm to segment different battery usage behaviors among EV users. While the study certainly has merits, I am concerned with the contributions of the study. As a study for the application of ML methods, I am not fully convinced of the practical importance of clustering EV drivers purely based on the timestamp and the SoC consumption, and how the obtained clustering insights could be used to promote social goods without location and contextual information. Moreover, the outcomes of the study are believed to be very sensitive to the feature engineering process that is problematic, which is mainly due to the quality of the data itself. Finally, the reviewer believes that for this type of application, the unique insights that are not easily obtainable through standard clustering approaches would be more important than the clustering quality quantified by the `Silhouette Score', especially when no ground truth information is available to validate what is the `optimal` number of clusters. The study currently does not include justifications on how the proposed method can uniquely extract important hidden information. The reviewer believes that the study is more suitable for publication in an interdisciplinary journal rather than ICRL.

**Strengths:**

The study uses a unique dataset that contains detailed information on electric vehicle dynamics from 3100 EV drivers.

**Weaknesses:**

The authors did not sufficiently defend their motivations and the reviewer is not convinced by the importance of the study.  Why is it important to cluster similar EV vehicles based on their battery consumption? What does this tell us? After reading through the paper, I still have a hard time connecting how such clustering would contribute to the grid management, without additional information on their location information.

The writing of the introduction section is also a bit disorganized. The authors started with existing efforts in clustering EV usage patterns and then suddenly jumped to GNNs in a completely different context. I did not find a strong reasoning as to why the proposed `cascading clustering` scheme is necessary for the proposed task. The current presentation left me with the impression that the author is trying to find nails with a hammer in hand.

The data used in this study is not of sufficient quality to support the aim of the authors, and the authors have to introduce predefined thresholds (feature engineering) to tell charging events from driving events and vehicles being idle. The introduced threshold (2% SOC change in 10 minutes) is problematic, and will likely count slow charging as driving events. Note that an hour of slow charging will yield a 1 kWh increase in SoC or equivalently ~2%, meaning that a slow charging, in this case, will provide fewer than 0.3% of SoC in 10 minutes. This point alone would totally destroy the outcomes of the study.

The results are more qualitative discussions that explain the contextual insights from the clustering outcomes. It would be more favorable if the results emphasized distinct insights that are only obtainable through the design of such a clustering algorithm and why such insights are important for practitioners.

**Questions:**

What are the fundamental contributions of the cascading clustering algorithm that is not obtainable through a typical hierarchical clustering method?

---

### Official Review · Reviewer_oWmf · 2023-10-30

**Soundness:** 2 fair
**Presentation:** 3 good
**Contribution:** 3 good
**Rating:** 5
**Confidence:** 4

**Summary:**

The authors study the battery usage of patterns of electric vehicles (EVs), by implementing a novel two-stage cascaded clustering approach.  In the first stage, the clustering analysis focuses on various factors, including state of charge, depth of discharge, and charging duration, uncovering unique battery utilization patterns associated with each cluster. In the second stage, based on these clusters, the authors use LSTM and GCN-based deep learning techniques for further analysis.  The two-stage method effectively extracts insights from temporal EV charging data, contributing to a deeper understanding of user charging profiles and behaviors. The method could potentially enhance EV system efficiency and personalization by considering the unique attributes of each identified cluster.

**Strengths:**

-	Innovative Method: The paper introduces an innovative cascaded clustering approach that combines time-series data analysis and deep learning techniques to reveal electric vehicle (EV) battery usage patterns. This approach is relatively novel and can provide valuable insights into EV battery management.
-	Interpretability: The paper provides detailed information about EV battery usage patterns, including charging speed, depth of discharge, and other factors, through visualizations.
-	Advanced Deep Learning Application: In the second stage of clustering, the authors introduce deep learning techniques, such as LSTM and Graph Convolutional Networks (GCN), to better analyze and classify vehicle usage patterns.
-	Social Effect: The importance of this research is highlighted by its practical implications for EV battery management, a pertinent topic considering the growing EV adoption. The study offers practical insights into EV battery usage patterns, making it a significant contribution beyond academia.

**Weaknesses:**

-	Necessity in Two-Stage Clustering: The necessity and practical implications of this two-stage clustering remain unclear. The paper includes extensive discussion and visualization on the first stage result. However, the first-stage clusters seem to have limited effectiveness on metrics, while the second stage, using advanced techniques, lacks a clear interpretation or downstream utility for the extracted information or hierarchical structures.
-	Quantitative Validation: While the paper presents quantitative results for clustering, it lacks quantitative validation for downstream tasks. Quantifying the impact of the method on these tasks would strengthen the paper's contributions and practical relevance.

**Questions:**

-	What advantages or unique insights does the cascaded clustering bring, and is there any empirical evidence supporting this necessity?
-	The authors mention the scalability challenges of raw clustering on their dataset. Does the method generalize well to different EV models or regions, or is it highly specific to the dataset used?